# Identifying the Psychometric Properties of the Malay Version of the WHOQOL-BREF among Employees with Obesity Problem

**DOI:** 10.3390/ijerph19127542

**Published:** 2022-06-20

**Authors:** Mohd Helma Rusyda, Nor Ba’yah Abdul Kadir, Wan Nur Khairunnisa Ismail, Siti Jamiaah Abdul Jalil, Nurul-Azza Abdullah, Arena Che Kasim, Suzana Mohd. Hoesni, Mohd Rizal Abdul Manaf

**Affiliations:** 1Centre for Research in Psychology and Human Well-being, Faculty of Social Sciences and Humanities, Universiti Kebangsaan Malaysia, Bangi 43600, Malaysia; aknbayah@ukm.edu.my (N.B.A.K.); nisaismail281@gmail.com (W.N.K.I.); nurulazza@ukm.edu.my (N.-A.A.); arena@ukm.edu.my (A.C.K.); smh@ukm.edu.my (S.M.H.); 2Department of Dakwah and Leadership, Faculty of Islamic Studies, Universiti Kebangsan Malaysia, Bangi 43600, Malaysia; sitijamiaah82@ukm.edu.my; 3Department of Community Health, Faculty of Medicine, Universiti Kebangsaan Malaysia, Bangi 43600, Malaysia; mrizal@ppukm.ukm.edu.my

**Keywords:** WHOQOL, quality of life, psychometric properties, back translation, Malay translation, obese employees

## Abstract

The Malay version of the WHOQOL-BREF was published approximately 15 years ago. Since then, no known research has been conducted to identify the psychometric properties of the scale using confirmatory factor analysis. This study aimed to establish a model by applying a scientific approach to the translation and adaptation method. The back translation technique was used for the translation process. This cross-sectional study involved 282 employees at Universiti Kebangsaan Malaysia. The instrument received satisfactory Cronbach’s alpha reliability values. The data were analysed with SEM using AMOS. Results showed that the model produced is parsimonious, with CMIN/df = 0.23, CFI = 0.93, SRMR = 0.08, RMSEA = 0.08 and PCLOSE = 0.07. Adopting the Malay version of the WHOQOL-BREF for future research is highly recommended due to its properties.

## 1. Introduction

The assessment of quality of life in relation to health has been well established worldwide [1,2,3]. The World Health Organization Quality of Life (WHOQOL) assessment is the most widely used self-reporting measure for examining the quality of life among the general population [1,4], patients [5,6], students [7,8], community [9,10] and employees [11,12]. The WHO defines quality of life as “an individual’s perception of their position in life in the context of the culture and value systems in which they live and in relation to their goals, expectations, standards and concerns” [13]. This instrument consists of 24 facets of QOL, each with 4 items, and 4 additional items relating to the “overall quality of life and general health”. The 24-item WHOQOL-BREF has been developed as a short version of the WHOQOL-100.

According to reports, the WHOQOL can be used across cultures [14,15,16] and has satisfactory psychometric properties. The short version, the WHOQOL-BREF, has been applied particularly in clinical settings and large-scale epidemiological studies. According to a systematic review of the research into quality of life in medicine and health sciences, more than 150 studies have used the WHOQOL-BREF and demonstrated acceptable psychometric results [17]. A bibliometric analysis from 2000–2019 showed that the main contributions to studies into quality of life came from North America and Europe, with fewer research from regions such as Asia, South America and Africa [18].

The earliest report on the psychometric properties of the Malay version of the WHOQOL-BREF was published in 2003 [19]. For the study sample, 200 participants were recruited from the physicians and psychiatric clinics at the Universiti Sains Malaysia Hospital. The sample consisted of 40 healthy participants and 160 patients with hypertension, diabetes mellitus, epilepsy or schizophrenia. The statistical analysis methods employed to report on the psychometric properties comprised the internal consistency, test–retest reliability, concurrent validity, criterion validity and discriminant validity methods. An exploratory factor analysis was used to assess the construct validity of the WHOQOL-BREF. A study [20] using a sample of disabled students found satisfactory Cronbach’s alpha values for the domains of the WHOQOL-BREF covering physical health (0.72), psychological health (0.82) and environmental health (0.80), whereas social relationships received a value of 0.69. The factor analysis found that the social relationship items had poor factor loadings (<0.40), indicating that some items were unsuitable for measuring the social relationships of disabled students. Similar findings were reported in a sample of disabled individuals in Malaysia [21].

To date, although the Malay version of the WHOQOL has been used in a Malaysian setting for 15 years, no studies have used a confirmatory factor analysis to validate it. Therefore, this study collected recent data from employees with obesity in Malaysia to assess the psychometric properties of the Malay version of the WHOQOL-BREF using a confirmatory factor analysis (CFA). The primary aim was to examine the construct validity and reliability of the Malay version of the WHOQOL-BREF. The study addressed the following research question: does the four-factor model of the WHOQOL-BREF show a satisfactory construct validity in terms of dimensionality, convergent and discriminant validity, and reliability (internal consistency, floor-ceiling) in university employees with obesity?

## 2. Materials and Methods

### 2.1. Participants

This study, which used a cross-sectional research design, included 282 participants (198 females, 84 males), who were employed at Universiti Kebangsaan Malaysia in Bangi, Malaysia. The participants varied in terms of their designation and level of employment and were defined as obese, as their Body Mass Index (BMI) was >25. The participants completed a self-reporting questionnaire via Google Forms.

### 2.2. Translation and Cultural Adaptation Methods

It is very important to choose the most appropriate approach and method for translation and adaptation [22]. This research employed the linguistic approach to translation and selected back-translation as the adaptation method. Linguistic translation is a grammar-focused translation, which provides an equivalent meaning between the source language (SL) and target language (TL) with similar psychometric properties, in relation to cultural, social and political contexts [23,24,25]. Compared to other translation approaches, linguistic translation provides equivalence, which may be impossible to achieve with a literal translation [26]. Equivalence occurs when two instruments that are assessing the same construct are compared and are confirmed to be valid [27,28,29,30], free from cultural bias, and are acceptable and relevant in the culture [31,32]. To achieve instrument equivalence, a researcher must choose the most appropriate method for translation. Therefore, back-translation was chosen for this study as it can produce equivalence for the research instruments, and consequently, for the research findings [33,34,35]. Equivalence is assessed through a detailed examination of the accuracy of an instrument produced by multiple translators and translations (extra checks) [36]. Back-translation is a three-step process, which involves translating to the TL, translating back to the SL, and comparing the versions [37]. The instrument must first be translated from the SL (English) into the TL (Malay language) by a bilingual translator. Next, a blind translator must translate the instrument back into the SL. All incongruities must be discussed between the translators [38,39,40]. To ensure the accuracy of the instrument, the translators were selected and the translation was conducted according to the guidelines [41] provided by Brislin. These guidelines include a formal qualification as a translator into the TL, first-language experience in the TL, excellent knowledge of English, experience of living and working in an English-language environment, and familiarity with the culture associated with the TL. Brislin originally recommended a 7-step process for back-translation [42].


**Step 1**


In this step, a document or measuring tool is translated to a TL. Researchers can translate a measuring tool if they are native speakers of the TL [43]. However, they must recognise that their formal education may influence the translation as they may have a propensity to use academic terms that are not well understood by potential participants [44]. Additionally, the researchers and participants may have a different understanding of the terms, and this will affect the congruency of the translation [45]. For this study, it was decided to translate the instrument into the Malay language (TL), and to then send the translated material to another translator. This will be discussed in Step 2.


**Step 2**


To overcome problems that may arise in Step 1, the translated instrument is sent to another translator, designated as Translator A (TA). In this step, the TA must translate the instrument back into the SL. Furthermore, a second translator (Translator B; TB) is given the original version in the SL to be translated to the TL. The main criteria for choosing the translators for this process are their qualifications and expertise in both languages [46]. Ideally, the translators should have experience in the SL and TL (English and Malay language). It should be noted that the translators for Step 2 should not have knowledge in the research field or subject because it can affect the meanings and terms [47]. On completion, the translated version in the SL (from TA) is submitted to the next translator.


**Step 3**


In this step, a translator must be knowledgeable in the SL and TL. Ideally, the translator should be qualified in the research field and the instrument [48]. Furthermore, it is highly recommended that the translator should have a high academic qualification, such as a master’s degree or doctoral candidate status [49].


**Step 4**


Next, discrepancies between versions of the instrument are discussed by translators. All the translated instruments and the original version in the SL are brought together and compared. In this step, the versions must be modified until the researcher and translators agree with the translation [50]. This stage is critical and time-consuming. Translators are encouraged to maintain the meanings despite having to make some minor changes during deliberations [51].


**Step 5**


After agreement has been reached on the translation result, the translated instrument can be distributed to potential respondents for pilot testing [52]. Although the number of respondents required for qualitative research is not specified, 10 respondents are typically regarded as adequate for a pilot test [53]. For quantitative research, at least 100 respondents are required to attain reliability and validity.


**Step 6**


In this step, all the data are gathered and analysed by the researcher. If the proposed model is shown to fit, then it can be said that the instrument demonstrates reliability and validity [53].


**Step 7**


If the researcher is satisfied with the findings, the instrument is used on the targeted population.

### 2.3. Measure

The 24-item WHOQOL-BREF is a self-reporting measurement that covers four areas: physical health (7 items), psychological health (6 items), social relationships (3 items) and environmental health (8 items). Participants respond to the items on a five-point Likert-type scale. The physical health domain includes items on mobility, daily activities, functional capacity, energy, pain and sleep. The psychological domain items include self-image, negative thoughts, positive attitudes, self-esteem, mentality, learning ability, memory concentration, religion and mental status. The social relationships domain contains questions on personal relationships, social support and sex life. The environmental health domain covers issues related to financial resources, safety, health and social services, physical living environment, opportunities to acquire new skills and knowledge, recreation, general environment (noise, air pollution, etc.) and transportation. The report showed that the WHOQOL-BREF has good psychometric properties.

### 2.4. Statistical Analysis

The participants completed self-reporting measurements via Google Forms. The responses were analysed with structural equation modelling-AMOS (SEM-AMOS) to identify the relationships between the variables and determine the fit of the model. Although there are no guidelines specifying the optimal sample size for a factor analysis, the larger the sample size, the smaller the standard error [54,55]. Nevertheless, it is important to note that the aim of this study was to test the theoretical model, not the population model; therefore, a sample size of N ≥ 200 was deemed to be sufficient [56,57,58].

A confirmatory factor analysis (CFA), which is a form of factor analysis, is also a theory-driven analysis [59]. It can measure convergent and discriminant validity [60] and confirm a hypothesis by demonstrating an established relationship between observed variables and their underlying latent constructs [61]. An SEM-AMOS analysis was used in this study because of its ability to test a hypothesis based on a theoretically-specified model [62].

## 3. Results

To determine whether the sample size was sufficient, a KMO test was employed to examine the sampling adequacy for a factor analysis. The KMO measure of sampling adequacy standards are as follows: 0.00 to 0.49 (unacceptable), 0.50 to 0.59 (miserable), 0.60 to 0.69 (mediocre), 0.70 to 0.79 (middling), 0.80 to 0.89 (meritorious) and 0.90 to 1 (marvellous) [63]. A value between 0.8 and 1 reflects the best cut-off point and fit for a factor analysis [64]. A KMO value of 0.89 was obtained for this analysis, thereby deeming it as “meritorious” [65], whereas the Cronbach’s alpha value was 0.88. Table 1 shows the correlations between the items. All the items were retained since none of them were highly correlated.

### 3.1. Factor Loading and Communalities

Although a factor loading of 0.30 was acceptable [66], other aspects, such as communality, had to be considered. Communality is the shared variance reflected by the sum of the squared factor loading [67]. The factors with the lowest communality values were identified based on the data shown in Table 2. As suggested, any communality below 0.20 was eliminated.

Table 3 presented the vaidity findings of this paper. The convergent validity of each factor is estimated based on an average shared variance (AVE) > 0.5 [67]. AVE is the average amount of variance in observed variables that a latent construct can explain [68]. The discriminant validity emphasises the items’ capacity to be distinct from other factors other than their parent factor [68]. The maximum shared squared variance (MSV) is the benchmark for discriminant validity [69]. A factor is distinct if the MSV value is smaller than the AVE [70].

For reliability, most researchers determine Cronbach’s alpha readings as 0.70 and above [71]. However, Cronbach’s alpha value is the most basic reliability testing and can be used if a model has only one factor [72]. The most common measurements for reliability are composite reliability (CR), and maximal reliability (MaxR (H)) because these tests can accurately measure reliability [73]. CR reading is achieved when all the “standardized” items are allowed to correlate with each other (intraclass correlation). The benchmark in evaluating CR is >0.7 and, usually, MaxR (H) has a higher reading than CR [74]. Normality testing findings can be examined in Table 4.

### 3.2. Model Fit Assessment

For a model fit assessment, the Hu and Bentler [75] threshold or cut-off point, namely, the chi-square to df ratio (χ^2^/df), comparative fit index (CFI), goodness-of-fit index (GFI), adjusted goodness-of-fit index (AGFI), standardised root mean square residual (SRMR), root mean square error of approximations (RMSEA) and PCLOSE function can be used. However, as recommended, the GFI and AGFI were disregarded in this study as they are highly sensitive to sample size [76]. The cut-off point for model fit indicated in Table 5.

As expected, four domains were produced from the analysis, in line with the WHOQOL-BREF framework. However, the initial finding of the modelling at a CFI of 0.86 was unsatisfactory. Therefore, the factor loadings and communalities were examined, and it was decided to remove all the items with communalities below 0.20. Finally, the default model had an acceptable parsimonious model fit with CMIN/df = 0.23, CFI = 0.93, SRMR = 0.08, RMSEA = 0.08 and PCLOSE at 0.07. The measurement model is presented in Figure 1.

## 4. Discussion

This study attempted to identify the psychometric properties of the Malay version of the 24-item WHOQOL-BREF. The translation was conducted in accordance with the back-translation protocol. The participants were university employees with obesity, mostly with a BMI of 25 or higher. Although the CFA indicated a good model fit, it was necessary to address a few issues. In addition, inconsistent items were identified and had to be eventually deleted from the model.

The study found four domains of the WHOQOL that were consistent with previous research on the WHOQOL-BREF [77,78,79]. Although some earlier research indicated that, in some cases, the factor extraction of the WHOQOL-BREF could be a one-factor solution [80,81], the four-factor solution was deemed necessary for a better model fit, as the former has disadvantages in terms of inter-item correlations [82,83].

Table 2 shows the communalities of each item in the WHOQOL-BREF domains. In the physical health domain, several items with low communalities were identified. For example, the item, “How much do you need any medical treatment to function in your daily life?”, had the lowest communality of 0.13, thereby suggesting that the item could not be explained by the factor. Although the research sample was university employees with obesity, this result was consistent with previous research on physical health among patients with diabetes mellitus, where it was found that medical treatment was not the most significant predictor in maintaining quality of life as compared to psychological health and social relationships [84,85,86]. Two other items, “To what extent do you feel that physical pain prevents you from doing what you need to do?” and “How well are you able to get around?”, also had low communalities, thereby indicating that these concerns were insignificant for the research participants with obesity and could not be explained by the physical domain factor. This finding was also consistent with that of previous research [12,87]. Therefore, it was decided to exclude these three items with low communalities from the physical domain.

Within the psychological domain, the item, “How often do you have negative feelings such as blue mood, despair, anxiety, depression?”, had the lowest communality. One reason for this may have been the fact that the question was double-barrelled. It might have been difficult for the participants to respond to multiple mental health issues in one item, as double-barrelled questions are open to predisposed answers or misapprehensions [88,89,90]. Furthermore, as suggested by the WHO, any WHOQOL-BREF questions that are double-barrelled should be edited or removed to maintain the validity and reliability [91].

Among eastern societies such as Malaysia, questions about sexual issues are often considered taboo. The item, “How satisfied are you with sex life?”, had a low communality value of 0.18 among all the items. However, this finding was not unexpected, based on the large number of previous research on sexual issues within the same context [92]. For example, a Turkish researcher had to remove an item on sexuality and personal life due to the low factor loading, eventually producing a model that did not fit the population. In a study on the Iranian population, the Cronbach’s alpha was improved by removing items on sexual activities [93]. Similar results were demonstrated by studies on sexual issues within a Malaysian setting. For instance, items on sexual intentions and premarital sex among Malaysian youths were removed due to a very low factor loading [94] or weak factor correlation [95].

In the environmental domain, the item, “To what extent do you have the opportunity for leisure activities?”, had the lowest communality value due to the daily work demands of Malaysian employees [96,97], leading to the feasible argument that the participants may have little opportunity to participate in leisure activities. Other than this result, the research findings supported those of previous research in the environmental domain and quality of life, which suggested that the environmental domain is strongly influenced by a quality environment [98,99]. Furthermore, this finding was consistent with previous research that indicated that compared to the other items, this item is a weak predictor of quality of life [100,101].

## 5. Implications

A systematic, scientific translation protocol and statistical analysis showed that the Malay version of the WHOQOL-BREF has excellent psychometric properties. Furthermore, the model that was produced was deemed to be parsimonious. Therefore, it is recommended that this version by adopted for any future research. However, it is proposed that factors with communality values of less than 0.20 be eliminated to avoid any issues during the modelling process. It is highly recommended that any future research that wishes to use the original items should refine all the items to ensure better inter-item correlations.

## 6. Conclusions

The study results are crucial to confirm whether the four-factor solution model can be replicated in the Malaysian setting. Although some of the items were removed, the model fit was satisfactory, and the psychometric properties were presented.

## Figures and Tables

**Figure 1 ijerph-19-07542-f001:**
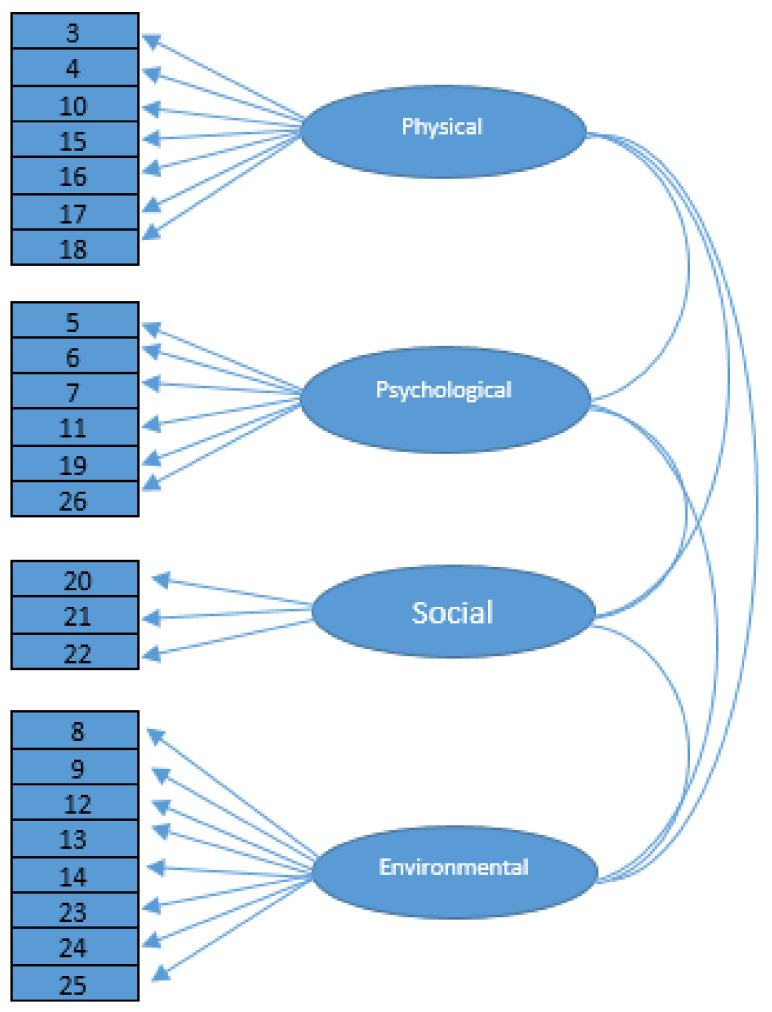
Measurement model analysis.

**Table 1 ijerph-19-07542-t001:** Correlations.

	1	2	3	4	5	6	7	8	9	10	11	12	13	14	15	16	17	18	19	20	21	22	23	24
1	1																							
2	0.48 **	1																						
3	−0.19 **	−0.15 *	1																					
4	−0.23 **	−0.21 **	0.71 **	1																				
5	−0.20 **	−0.21 **	0.54 **	0.59 **	1																			
6	−0.26 **	−0.16 *	0.46 **	0.58 **	0.65 **	1																		
7	−0.25 **	−0.13	0.33 **	0.44 **	0.40 **	0.61 **	1																	
8	−0.26 **	−0.22 **	0.39 **	0.44 **	0.46 **	0.52 **	0.46 **	1																
9	−0.16 *	−0.03	0.19 **	0.22 **	0.22 **	0.22 **	0.20 **	0.38 **	1															
10	−0.26 **	−0.11	0.32 **	0.34 **	0.33 **	0.48 **	0.36 **	0.41 **	0.45 **	1														
11	−0.25 **	−0.19 **	0.40 **	0.39 **	0.43 **	0.48 **	0.36 **	0.52 **	0.44 **	0.66 **	1													
12	−0.15 *	0.03	0.18 **	0.20 **	0.15 *	0.24 **	0.23 **	0.41 **	0.39 **	0.32 **	0.33 **	1												
13	−0.31 **	−0.30 **	0.25 **	0.26 **	0.19 **	0.34 **	0.35 **	0.45 **	0.23 **	0.26 **	0.33 **	0.39 **	1											
14	−0.17 **	−0.09	0.27 **	0.25 **	0.28 **	0.27 **	0.23 **	0.36 **	0.29 **	0.35 **	0.30 **	0.33 **	0.25 **	1										
15	−0.40 **	−0.20 **	0.36 **	0.35 **	0.32 **	0.41 **	0.39 **	0.60 **	0.33 **	0.40 **	0.40 **	0.42 **	0.47 **	0.58 **	1									
16	−0.24 **	−0.23 **	0.36 **	0.36 **	0.39 **	0.41 **	0.37 **	0.51 **	0.22 **	0.28 **	0.35 **	0.31 **	0.36 **	0.45 **	0.66 **	1								
17	−0.29 **	−0.16 *	0.40 **	0.51 **	0.43 **	0.46 **	0.39 **	0.42 **	0.48 **	0.38 **	0.41 **	0.27 **	0.33 **	0.42 **	0.57 **	0.59 **	1							
18	−0.23 **	−0.18 **	0.38 **	0.43 **	0.47 **	0.43 **	0.35 **	0.40 **	0.33 **	0.36 **	0.39 **	0.18 **	0.24 **	0.43 **	0.53 **	0.52 **	0.68 **	1						
19	−0.07	−0.06	0.14 *	0.20 **	0.25 **	0.27 **	0.24 **	0.19 **	0.19 **	0.27 **	0.18 **	0.13	0.19 **	0.23 **	0.27 **	0.32 **	0.36 **	0.59 **	1					
20	−0.20 **	−0.21 **	0.37 **	0.45 **	0.46 **	0.32 **	0.34 **	0.35 **	0.31 **	0.28 **	0.35 **	0.13 *	0.27 **	0.31 **	0.41 **	0.50 **	0.56 **	0.53 **	0.36 **	1				
21	−0.25 **	−0.18 **	0.28 **	0.37 **	0.40 **	0.40 **	0.35 **	0.29 **	0.21 **	0.42 **	0.36 **	0.13 *	0.26 **	0.33 **	0.35 **	0.37 **	0.45 **	0.56 **	0.44 **	0.57 **	1			
22	−0.13 *	−0.18 **	0.34 **	0.34 **	0.35 **	0.33 **	0.24 **	0.32 **	0.24 **	0.28 **	0.35 **	0.11	0.26 **	0.24 **	0.25 **	0.21 **	0.44 **	0.46 **	0.31 **	0.40 **	0.51 **	1		
23	−0.20 **	−0.25 **	0.30 **	0.36 **	0.31 **	0.35 **	0.25 **	0.30 **	0.15 *	0.33 **	0.31 **	0.1	0.25 **	0.25 **	0.29 **	0.35 **	0.45 **	0.40 **	0.32 **	0.41 **	0.57 **	0.58 **	1	
24	0.23 **	0.26 **	−0.23 **	−0.22 **	−0.23 **	−0.30 **	−0.21 **	−0.21 **	−0.16 *	−0.20 **	−0.17 *	−0.11	−0.24 **	−0.33 **	−0.29 **	−0.27 **	−0.38 **	−0.36 **	−0.19 **	−0.21 **	−0.26 **	−0.21 **	−0.16 *	1

** Correlation is significant at the 0.01 level (two-tailed). * Correlation is significant at the 0.05 level (two-tailed). N = 228.

**Table 2 ijerph-19-07542-t002:** Factor loading and communalities.

Factor	Item No.	Item	Factor Loading	Communality
	3	To what extent do you feel that physical pain prevents you from doing what you need to do?	0.41	0.17
**1**	4	How much do you need any medical treatment to function in your daily life?	0.36	0.13
**Physical**				
	10	Do you have enough energy for everyday life?	0.58	0.33
	15	How well are you able to get around?	0.40	0.16
	16	How satisfied are you with your sleep?	0.42	0.21
	17	How satisfied are you with your ability to perform your daily living activities?	0.69	0.47
	18	How satisfied are you with your capacity for work?	0.61	0.38
	5	How much do you enjoy life?	0.56	0.31
	6	To what extent do you feel your life to be meaningful?	0.64	0.44
**2**	7	How well are you able to concentrate?	0.59	0.34
**Psychological**	11	Are you able to accept your bodily appearance?	0.45	0.20
	19	How satisfied are you with yourself?	0.68	0.47
	26	How often do you have negative feelings such as blue mood, despair, anxiety, depression?	0.28	0.08
	20	How satisfied are you with your personal relationships?	0.68	0.46
**3**	21	How satisfied are you with sex life?	0.42	0.18
**Social**	22	How satisfied are you with the support you get from your friends?	0.53	0.28
	8	How safe do you feel in your daily life?	0.66	0.44
	9	How healthy is your physical environment?	0.50	0.25
**4**	12	Have you enough money to meet your needs?		
**Environmental**	13	How available to you is the information that you need in your day-to-day life?	0.57	0.32
	14	To what extent do you have the opportunity for leisure activities?	0.36	0.13
	23	How satisfied are you with the conditions of your living place?	0.57	0.33
	24	How satisfied are you with your access to health services?	0.48	0.23
	25	How satisfied are you with your transport?	0.52	0.27

**Table 3 ijerph-19-07542-t003:** Convergent and discriminant validity.

Factors	CR	AVE	MSV	Max R(H)	AC
Physical	0.75	0.53	0.10	0.97	0.71
Psychological	0.90	0.60	0.15	0.94	0.77
Social	0.83	0.62	0.07	0.97	0.79
Environmental	0.78	0.55	0.14	0.83	0.74

N = 228. CR = composite reliability, AVE = average variance extracted, MSV = maximum shared variance, Max R (H) = maximum reliability, AC = Cronbach’s alpha.

**Table 4 ijerph-19-07542-t004:** Normality test findings.

Factor	Item No.	Item	M	SD	Skewness	Kurtosis	Range
	3	To what extent do you feel that physical pain prevents you from doing what you need to do?	2.77	0.85	0.01	0.17	1–5
**1** **Physical**	4	How much do you need any medical treatment to function in your daily life?	2.18	0.90	0.33	−0.31	1–5
	10	Do you have enough energy for everyday life?	3.77	0.71	−0.00	0.40	2–5
	15	How well are you able to get around?	4.01	0.95	−0.94	0.67	1–5
	16	How satisfied are you with your sleep?	3.50	0.83	−0.35	−0.35	1–5
	17	How satisfied are you with your ability to perform your daily living activities?	3.64	0.76	−0.76	1.17	1–5
	18	How satisfied are you with your capacity for work?	3.79	0.68	−0.94	2.40	1–5
	5	How much do you enjoy life?	3.81	0.64	0.10	−0.45	2–5
	6	To what extent do you feel your life to be meaningful?	4.07	0.69	0.17	−0.61	2–5
**2**	7	How well are you able to concentrate?	3.70	0.63	0.13	−0.40	2–5
**Psychological**	11	Are you able to accept your bodily appearance?	3.38	1.11	−0.00	−0.57	1–5
	19	How satisfied are you with yourself?	3.75	0.72	−0.51	10.12	1–5
	26	How often do you have negative feelings such as blue mood, despair, anxiety, depression?	2.24	0.61	1.5	3.16	1–5
	20	How satisfied are you with your personal relationships?	3.85	0.75	−0.96	20.18	1–5
**3**	21	How satisfied are you with sex life?	3.81	0.75	−1.02	2.51	1–5
**Social**	22	How satisfied are you with the support you get from your friends?	3.88	0.67	−0.38	0.95	1–5
	8	How safe do you feel in your daily life?	3.68	0.65	0.23	−0.49	2–5
	9	How healthy is your physical environment?	3.46	0.66	0.10	0.39	1–5
**4**	12	Have you enough money to meet your needs?	3.52	0.88	0.01	−0.35	1–5
**Environmental**	13	How available to you is the information that you need in your day-to-day life?	3.72	0.68	−0.03	−0.22	2–5
	14	To what extent do you have the opportunity for leisure activities?	3.26	0.93	0.94	−0.17	1–5
	23	How satisfied are you with the conditions of your living place?	3.99	0.64	−0.30	0.41	2–5
	24	How satisfied are you with your access to health services?	3.96	0.66	−0.42	1.17	1–5
	25	How satisfied are you with your transport?	4.07	0.58	−0.14	0.50	2–5

**Table 5 ijerph-19-07542-t005:** Cut-off point for model fit.

Measure	Threshold Value
Chi-square/df (CMIN/df)	<3 good
CFI	>0.95 great; >0.90 acceptable
SRMR	<0.09
RMSEA	<0.05 good; 0.05–0.10 moderate; >0.10 bad
PCLOSE	>0.05

## Data Availability

The data presented in this study are available within the article.

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
