# Peer review of "Identifying the Psychometric Properties of the Malay Version of the WHOQOL-BREF among Employees with Obesity Problem"

_ijerph, 2022, doi:10.3390/ijerph19127542_

Round 1

Reviewer 1 Report

Thank you very much for allowing me to review your work.

I have some questions that I would like if you can clarify:

-          In line 151 there is talk about "for both populations" what populations are you referring to?

-          In line 156 "if the researcher is satisfied with the findings" why are the results obtained in that pilot not described?

-          It is not clear to me how convergent validity is measured. What gold standard has been used? And what are the results?

-          How was the ceiling-floor effect studied? And what results were obtained?

-          Was test-retest reliability studied?

-          Cronbach`s alpha value, was it studied for each dimension? Why isn't a table with that data included?

-          Considering the implications, why is a proposal not made by eliminating the items that are not applicable?

Author Response

1) In line 151 there is talk about "for both populations" what populations are you referring to?

I have deleted the statement. 

2) In line 156 "if the researcher is satisfied with the findings" why are the results obtained in that pilot not described?

We presumed this article is the pilot study of our whole research project. 

Our project involved one pilot study (this article), and an experimental longitudinal research design.

We used the established items from this manuscript for the longitudinal research design. 

We tested CFA for the longitudinal research data and the model is parsimonious and has an acceptable model fit.

3) It is not clear to me how convergent validity is measured. What gold standard has been used? And what are the results?

We have included the convergent and discriminant validity

4) How was the ceiling-floor effect studied? And what results were obtained?

We have added the normality findings in the article

5) Was test-retest reliability studied?

No test-retest reliability was conducted because this article is presumed to be the pilot study of our research project.

6) Cronbach`s alpha value, was it studied for each dimension? Why isn't a table with that data included

We have included the data in convergent and discriminant validity.

7) Considering the implications, why is a proposal not made by eliminating the items that are not applicable?

As a continuation of this (pilot) research, we conducted longitudinal research. In the longitudinal research, we disregarded items below .20.

Reviewer 2 Report

Thank you very much for the opportunity to review the manuscript entitled " Identifying the psychometric properties of the Malay version of the WHOQOL-BREF among employees with obesity problem", which was sent to IJERPH.

This research fits into the main topic of the journal.

Despite the fact that this topic is interesting, and I really think that the authors have devoted a lot of effort to their research, this current manuscript still needs improvement.

Since the SEM method is sensitive to the distribution of variables, it is necessary to provide data on the distribution parameters of each questionnaire item, as well as on the scales obtained as a result of the confirmatory analysis. In addition, it would be appropriate to also provide Alpha data for each scale.

It is also necessary to provide data on the design validity of the tool.

Finally, for the practical use of the methodology, it is necessary to standardize it.

 This manuscript holds actual value to the readers on IJERPH.

I will be glad to review the revised manuscript.

Author Response

Since the SEM method is sensitive to the distribution of variables, it is necessary to provide data on the distribution parameters of each questionnaire item, as well as on the scales obtained as a result of the confirmatory analysis. In addition, it would be appropriate to also provide Alpha data for each scale.

-We have included the normality test findings in the manuscript

It is also necessary to provide data on the design validity of the tool.

-We have added the validity table in the manuscript

Round 2

Reviewer 1 Report

I consider that the article can be accepted in its current format.

Reviewer 2 Report

Many thanks for the revision of the manuscript. The article has become clearer and clearer. The data obtained can be compared with the results of other studies.